# Broadband Measurements of Soil Complex Permittivity

**DOI:** 10.3390/s23115357

**Published:** 2023-06-05

**Authors:** Justin Stellini, Lourdes Farrugia, Iman Farhat, Julian Bonello, Raffaele Persico, Anthony Sacco, Kyle Spiteri, Charles V. Sammut

**Affiliations:** 1Department of Physics, University of Malta, MSD2080 Msida, Malta; 2Department of Environmental Engineering DIAM, University of Calabria, 87036 Cosenza, Italy; 3Institute of Earth Systems, University of Malta, MSD2080 Msida, Malta

**Keywords:** dielectric constant, soil water content, water conservation

## Abstract

Agriculture is a major consumer of freshwater and is often associated with low water productivity. To prevent drought, farmers tend to over-irrigate, putting a strain on the ever-depleting groundwater resources. To improve modern agricultural techniques and conserve water, quick and accurate estimates of soil water content (SWC) should be made, and irrigation timed correctly in order to optimize crop yield and water use. In this study, soil samples common to the Maltese Islands having different clay, sand, and silt contents were, primarily, investigated to: (a) deduce whether the dielectric constant can be considered as a viable indicator of the SWC for the soils of Malta; (b) determine how soil compaction affects the dielectric constant measurements; and (c) to create calibration curves to directly relate the dielectric constant and the SWC for two different soil types of low and high density. The measurements, which were carried out in the X-band, were facilitated by an experimental setup comprising a two-port Vector Network Analyzer (VNA) connected to a rectangular waveguide system. From data analysis, it was found that for each soil investigated, the dielectric constant increases notably with an increase in both density and SWC. Our findings are expected to aid in future numerical analysis and simulations aimed at developing low-cost, minimally invasive Microwave (MW) systems for localized SWC sensing, and hence, in agricultural water conservation. However, it should be noted that a statistically significant relationship between soil texture and the dielectric constant could not be determined at this stage.

## 1. Introduction

Despite its crucial importance, the agricultural sector is often perceived as a primary contributor to freshwater demand [1]. According to the International Water Management Institute (IWMI), about 70% of the planet’s freshwater fraction diverted for human needs is currently being used to sustain the agricultural sector [2,3]. Additionally, the net evapotranspiration from global agricultural land might experience a twofold increase in the next fifty years if the current food consumption trends and agricultural practices are not revised immediately [4]. Past studies have shown repeatedly that the topic of water-use efficiency, especially agricultural water productivity, must be taken more seriously [4,5]. If immediate action is not taken on a global scale, we could be contributing to an already-brewing geopolitical ‘water war’ that could jeopardize the global economic system.

Typically, farmers wish to avoid drought stress throughout the growing season as it can instantly reduce the quantity and quality of crop yield. Applying “a little extra” irrigation is generally perceived as an easy insurance against these problems. However, this could be deleterious to the crop if the Soil Water Content (SWC) in the active root zone reaches and exceeds the field capacity. The latter defines the upper limit of the soil’s water storage for crop use. Any amount of water applied beyond this limit begins to immediately drain by gravity out of the root zone, is lost for crop use, and leaches nitrates, a valuable nutrient, and a groundwater contaminant [6,7]. Excess SWC also promotes root diseases and can severely shorten the productive life of trees [8]. Therefore, over-irrigation is not just considered a waste of water resources but also causes crop degradation that will be significant, especially for small-scale farms.

The enhancement of irrigation water management in the agricultural sector necessitates improved modern technologies whereby quick and accurate measurements of the water content in a particular soil type can be made. There are numerous methods that can be used for SWC quantification, with the traditional gravimetric methods maintaining their role as referential standards. However, methods that benefit from the soil’s dielectric properties, and perhaps specifically from the soil complex permittivity, have been gaining importance in recent years. By definition, the complex permittivity, $\varepsilon_{r}{}^{*}$, is a complex quantity that can be represented by the following expression:(1)$$\varepsilon_{r}{}^{*}=\varepsilon_{r}\prime -j\varepsilon_{r}\prime \prime$$
where $\varepsilon_{r}\prime$ is the dielectric constant, which corresponds with energy storage, $j=\sqrt{-1}$, and $\varepsilon_{r}\prime \prime$ is the loss factor associated with energy losses contributed by ionic drift and relaxation phenomena.

Several studies on the characterization of the link between complex permittivity, soil texture (determined by the sand, silt, and clay contents), and the SWC in the microwave (MW) regime, have been reported during the past forty years or so (see for example: [9,10,11,12,13,14,15,16,17,18,19,20,21]). Some of the most commonly used methods are free-space transmission techniques and transmission methods on infinite lines, coaxial line procedures, and Time Domain Reflectometry (TDR). Conflicting observations have been reported in this regard. For instance, the results reported by [11,22] suggest that the soil texture has a very weak influence on the dielectric constant of wet soil. Conversely, from other studies, e.g., [13,16,20], it was concluded that different soil types containing the same volumetric water content result in different magnitudes of $\varepsilon_{r}{}^{*}$. Given the high variability in the types of samples used during measurements, sample preparation methods, test frequency bands, and measurement methods amongst the different studies, it is relatively difficult to deduce the exact cause of these differences.

It is generally agreed that in the MW regime, both $\varepsilon_{r}\prime$ and $\varepsilon_{r}\prime \prime$ tend to increase with increased SWC (especially after the transition moisture value is exceeded). However, this relationship has never been investigated for the soils of Malta. Thus, the primary aim of our investigation was to deduce whether $\varepsilon_{r}\prime$ can be considered as a viable SWC indicator for the soils of Malta and establish a correlation between SWC and $\varepsilon_{r}\prime$ at different densities. Additionally, a correlation between the dielectric constant and SWC was established, characterizing the typical environment of soil in the field. In order to obtain an accurate characterization, the relevant parameters (SWC, soil type, density, etc.) were set under a controlled environment, and thus, a small-scale analysis was carried out in a lab using an existing rectangular waveguide method incorporating different conversion algorithms. Such correlation data facilitate the development process of minimally invasive and low-cost MW applications for localized SWC sensing, such as the one outlined in [23]. Thus, they can serve as the basis for numerical and analytical models to retrieve the SWC from measurements of the dielectric constant.

In total, six soil types common to the Maltese Islands, which will be described at a later stage, were investigated. It should be noted that from this point onwards, $\varepsilon_{r}\prime$ and $\varepsilon_{r}\prime \prime$ are simply referred to as ε′ and ε″ for simplicity.

The paper is structured as follows. In the next section, the experimental setup used is first discussed and then the materials used during the measurements, including materials used for the setup validation as well as soil samples, are described. Subsequently, the procedures adopted during the measurement sessions are explained. In Section 3, plots exhibiting outcomes from the different measurement procedures are presented together with some observations and interpretations. Finally, the significance of our results and the required future improvements are pointed out in Section 4. 

## 2. Materials and Methods

### 2.1. Experimental Setup and Procedure

Our measurements were facilitated by an experimental setup comprising the two-port Rohde and Schwarz ZVA-50 10 MHz–50 GHz Vector Network Analyzer (VNA) connected to an X-band rectangular waveguide system enabling measurements from 8.2 to 12.4 GHz. A three-dimensional model of the waveguide setup is shown in Figure 1, while the dimensions of the respective sections are provided in Figure 2. The VNA, which comprises a signal generator and a set of receivers, measures and outputs a set of reflection and transmission parameters (four in total for a two-port network). These coefficients, commonly known as S-parameters, can then be converted to complex permittivity and permeability values through different mathematical algorithms. 

In this study, Keysight Agilent 85071 measurement software, which was linked directly to the VNA, was used. This package included a set of conversion algorithms from which three, namely the Nicolson–Ross–Weir (NRW) [24,25], Epsilon, or NIST, Precision (EP) [26], and Polynomial Fit (PF) [27], were chosen for initial testing. 

The experimental procedure was divided into four main parts: (a) validation of the experimental setup, which was performed to: (i) ensure that the proposed experimental setup was well calibrated; (ii) identify the ideal conversion algorithm for the specific Material Under Test (MUT) in this study; and (iii) determine whether a layer of Kapton tape, which was meant to hold the soil samples in place, affects the final permittivity measurements, and if so, to what extent; (b) density measurements to determine the relationship between soil compaction and $\varepsilon_{r}\prime$; (c) moisture measurements to deduce whether the relationship between SWC and $\varepsilon_{r}\prime$ at different MW frequencies is statistically significant; and (d) to generate two cubic calibration models relating SWC with $\varepsilon_{r}\prime$ for two different soil types (Calcisol and Leptosol) at different densities.

### 2.2. Standard Materials

The dielectric materials Flame Retardant 4 (FR4), Polytetrafluorethylene (PTFE or Teflon), and 30% Glass Fiber Reinforced Polyethylene (hereinafter referred to as PE-30), which are three of the most common dielectrics used for low-loss electrical applications, were used for the validation procedure. These materials have come to be perceived as referential standards, primarily due to the vast body of available literature concerning their dielectric properties.

### 2.3. Soil Samples

In this study, six soil samples common to the Maltese Islands, here referred to as Bajjad 3 and 2 (B3 and B2), Garigue (G1 and G5), and Ħamri 5 and 9 (Ħ5 and Ħ9), were used. These were all provided by the Institute of Earth Systems of the University of Malta, Msida, Malta. The soils of Malta are considered to be relatively young and immature and in some instances have characteristics that are very similar to those of the parent material [28]. The Maltese Islands are characterized by a limestone sedimentary environment [29] and thus, the soil is highly calcareous. According to a number of studies, such as [30,31,32,33], the development and characteristics of the soils of Malta are not significantly influenced by the climate but are greatly affected by the nature of the parent material and human activity, especially in highly cultivated regions. In 1960, Lang [30] laid out the primary foundation for the understanding of these soils and their development. Through his study, he classified the soils of the island into three different categories using the Kubiëna classification system [34]: Carbonate Raw, Xerorendzina, and Terra soils. In 2003, the soil was reclassified using the World Reference Base for Soil (WRBS) [35]; however, as Lang’s system of classification is still widely used locally, it is appropriate to describe the three main soil categories proposed by Lang.

The Carbonate Raw soils, containing a calcium carbonate content of 80–90% and low levels of organic matter (OM), are further subdivided into four series: two primarily formed from blue clay parent material, one from weathered greensand, and another from dune sand. The Xerorendzinas are subdivided into three series that are mainly composed of a globigerina limestone parent material. These soils, which tend to have a greyish color and a powdery texture when dry, are generally characterized by a high chalk and gypsum content and limited organic matter (although they contain more than the carbonate soils discussed previously). Lastly, the Terra soils, famous for their reddish color (resulting from a high iron oxide content) and high fertility, may occur as Terra Fusca or Terra Rossa (both derived from upper and lower coralline limestone and lower globigerina parent material). Their carbonate content ranges from 2 to 15%, and the organic matter is generally higher than in the other two soil types. Using the latest WRBS classification system, the soils are classified as Calcisols, which comprise around 22% of the soils of the island, Regosols, Cambisols, Luvisols, Leptosols, Vertisols, and Arenosols. The Carbonate Raw Soils are mainly Regosols, Arenosols, Calcisols, and Vertisols; the Xerorenzinas are Calcisols, Cambisols, and Luvisols; and the Terra soils are Luvisols and Leptosols.

The textural characteristics, organic matter contents, and salinity of the soil samples used in this study are presented in Table 1. The salinity values are those of the saturated extract (EC_(SE)_). This was measured in the water that was extracted by suction from a soil water-saturated paste after 16 h incubation. The organic matter of the soil was determined using the Walkley and Black method [36] without additional heating, and the textural characteristics were determined using the hydrometer method [37].

### 2.4. Validation Procedure

The first step prior to conducting soil measurements was to carry out some validation measurements. Three conversion algorithms (NRW, EP, and PF) were tested using three reference dielectric materials, namely FR4, PTFE, and PE-30. Given the unconsolidated nature of the soil samples, something had to be used to contain them in the sample holder for the VNA measurements. In this case, Kapton tape was used, as shown in Figure 3, and thus, during the validation procedure, the possible effect of this layer attached to the two faces of the sample holder on the permittivity measurements was investigated. From this point onwards, the term ‘covered measurements’ is used when referring to scenarios in which the samples were covered with Kapton tape. By extension, the term ‘uncovered measurements’ is used when no Kapton tape was used. The validation measurements were distributed over six Through, Reflect, Line (TRL) calibrations. Throughout the entirety of the measurement procedure, the frequency range was set from 8.2 to 12.4 GHz since this is the recommended frequency range for fundamental mode propagation in this waveguide, which has a cut-off frequency of 6.5 GHz. 

Firstly, the uncovered measurements were considered. One by one, the sample holders containing the different materials were placed between the waveguide sections, and the respective system components were tightened accordingly. For each sample, a total of three measurements were carried out, using a different conversion algorithm each time. Once the measurements of the uncovered samples were completed, small sections of Kapton tape were cut and attached to the three sample holders in preparation for the set of covered measurements. However, before initiating the measurements, a new TRL calibration, this time with Kapton tape attached to the ends of the two waveguide sections, had to be performed. After the calibration, the actual measurements on the covered samples were initiated and once again, the three conversion algorithms were tested for all covered materials.

### 2.5. Density Measurements

Wet density measurements were carried out to analyze the relationship between the compaction of the soil samples in the sample holder and the resulting permittivity values. These were performed using all six soil types. For each soil type, three wet density levels (low, medium, and high) were considered, and two separate calibrations each consisting of two repeated measurements for each density level and conversion algorithm were carried out, yielding thirty-six measurements.

The density measurements were preceded by a ‘covered’ TRL calibration, which is a normal TRL calibration whereby a layer of Kapton tape is attached to the faces of the waveguide sections in contact with the calibration standards. To vary the density levels, the soil volume was kept constant (the entire volume of the sample holder gap) throughout the entirety of the measurement phase, but the soil was compressed to different degrees. The compression was done manually using a small lab spatula and the degree of compression was judged based on the net weight measured by the NBL 623i Nimbus Precision Balance having a resolution of 0.001 g. For low-density measurements, the soil samples were simply placed in the sample holder without being compressed; for medium density, the samples were compressed slightly after adding a small amount of soil; and, for the high-density cases, another small amount of soil was added and compressed further, making sure to keep the volume constant at all times. In carrying out this procedure, care was taken to ensure that consistency across different calibrations was maintained to eliminate bias and maximize correspondence. On average, the low-, medium-, and high-density values achieved were $1.321\pm 0.048{\;g\;\mathrm{cm}}^{-3}$, $1.496\pm 0.021{\;g\;\mathrm{cm}}^{-3}$, and $1.7184\pm 0.027{\;g\;\mathrm{cm}}^{-3}$, respectively. The density values reported here are quoted alongside their standard deviation. All density values obtained for the different soil samples as per VNA calibration are provided in Table 2.

### 2.6. Moisture Measurements

Moisture measurements were carried out on two of the soil samples, namely B3 and G1. In this case, seven moisture levels and two density levels (low and high for each moisture level) were considered. In total, two TRL calibrations were performed for each soil sample. Additionally, after each calibration, two repeated measurements were carried out for multiple combinations comprising different soil types, moisture content, density level, and conversion algorithm (in this case, the PF and EP conversion algorithms were tested). 

Adding water to the soil samples inevitably increases the soil’s wet density. During the moisture measurements, the primary aim was to analyze how the amount of water in the soil samples affects the permittivity. Thus, effects resulting from wet density changes had to be minimized when testing samples with different water content. Thus, care was taken to ensure that the volume of our sample holder always contained the same mass when doing the moisture measurements. This was done by varying the extent of compression to keep the wet density in the sample holder constant while varying the gravimetric water content. It should be noted that the drawback of this approach is that, when water is added to the soil, the soil particles accumulate together, and one does not have to compress the soil as much to fit it in the waveguide’s sample holder. Consequently, in higher saturation situations, the mass of water would represent a large portion of the total mass such that there would be fewer soil particles in the sample holder. This inevitably creates a form of bias given that in lower saturation situations, there will be more soil particles in the holder.

In order to conduct dielectric measurements as a function of moisture content, the following procedure was adopted. Initially, a ‘covered’ TRL calibration was carried out, and then, seven trays were each filled with approximately the same amount of soil of the same type. Different amounts of deionized water were added to the respective trays using an adjustable volume micropipette to achieve different SWC percentages. The wetted soil samples were then mixed well until no unsaturated, or otherwise oversaturated, patches remained. It should be noted that the saturation was limited to ~20%, given that beyond this point the soil acquired a paste-like texture such that the amount of compression could not be controlled. 

To determine the gravimetric soil water content ($\theta_{\mathrm{grav}}$) as a percentage contributed by the addition of water, the KERN DBS moisture analyzer, which is equipped with a 400 W Halogen quartz glass heater, was employed. In this study, the gravimetric water content was used instead of the volumetric as it better replicates measurements in the field. Moisture analyzers are ‘all-in-one’ instruments that can measure SWC using the ‘Loss on Drying’ (LOD) technique. A summary of the achieved gravimetric SWC percentages for B3 and G1, together with the corresponding density values, is presented in Table 3. In this case, ${\bar{\theta }}_{\mathrm{grav}}$, ${\bar{\rho }}_{\mathrm{low}}$, and ${\bar{\rho }}_{\mathrm{high}}$ refer to the average gravimetric SWC, low density, and high density, respectively. Once the moisture analyzer measurements were completed, the system calibration and permittivity measurements were initiated.

## 3. Results and Discussion

### 3.1. Conversion Algorithm Comparison

Table 4 shows a brief summary of the dielectric constant values obtained for the three reference materials when they were uncovered and for the three respective conversion algorithms. In this case, $\bar{\varepsilon \prime }$ signifies average values obtained from the entire frequency spectrum. Each computed average is accompanied by its corresponding standard deviation. Overall, the resulting values corroborate with values presented in other studies. As an example, referenced dielectric constant values range between 4.4 and 4.6 for FR4 [38,39,40,41], 1.9 and 2.1 for PTFE (Teflon) [38,42,43,44], and 2.7 and 2.8 for PE-30 [45]. Figure 4a–c depict ε′ as a function of frequency for the three reference materials, and the three differently colored profiles in each graph pertain to the chosen conversion algorithms. The standard deviation from the mean at each frequency point is exhibited as a shaded area above and below the respective profiles. Although these shaded areas might seem very significant in some instances, one must not forget that the scale on the y-axis is rather small. In fact, we observed that the standard deviation does not exceed 0.5.

Figure 4a exhibits the dielectric measurements of FR4 retrieved using three different algorithms parallel to each other. In this case, the three algorithms yielded a net average ε′ of 4.492. All profiles demonstrate an overall decrease in ε′ with increasing frequency, with the NRW and the PF profiles featuring the greatest and smallest variation in values, respectively. Consequently, the minimum and maximum values resulting from the PF method were the least dispersed from the mean. Cyclic oscillations can be seen in both the NRW and EP profiles. These oscillations are characterized by a series of peaks and troughs that increase with frequency. Despite this similarity, the NRW profile appears to be slightly right-shifted in comparison to that of the EP method, and its oscillations are less defined, especially at lower frequencies. Conversely, the resulting PF profile is relatively stable and its descent is not interrupted by any oscillations along the spectrum. The standard deviation for all profiles is largest at the lower frequencies (especially for the EP profile) and decreases gradually along the spectrum.

Figure 4b shows the inversion of the relative dielectric constant for PTFE, from the measured S-parameters. The resulting net average ε′ for this material is 2.035. In this case, neither of the profiles features the cyclic oscillations that were observed in the previously discussed scenario. The PF and EP algorithms yielded profiles of a very similar shape (relatively smooth with an almost unnoticeable overall decrease in ε′ with increasing frequency). In contrast, the NRW profile is characterized by small, chaotic, and irregular undulations at the lower frequencies, and two sharp dips at approximately 10.8 and 11.6 GHz (the latter being much more prominent). It can be observed that the largest range between the minimum and maximum ε′ is exhibited by the NRW profile (this is also highlighted by the large standard deviation in Table 4). The standard deviation shading suggests that, overall, the individual values returned by the EP and PF methods are the least dispersed from the corresponding means. The NRW standard deviation is greatest close to the observed dip at ~11.6 GHz.

Figure 4c illustrates the results obtained for PE-30. In this case, the resulting net average $\left(\varepsilon \prime \right)$ for the material is 2.685. The EP and PF dielectric profiles are very similar to those of PTFE. Despite this, their corresponding average values are slightly different. The NRW profile in this graph is also relatively similar in shape to that observed in the graph of PTFE, in that they both feature continuous irregular undulations along the frequency spectrum and occasional sharp variations; however, in the case of PE-30, the profile features a rapid increase just before plummeting down to the minimum resulting value. Although this latter behavior is not observed in the case of PTFE, the standard deviation profile just before the descent in Figure 4b mimics the behavior observed for PE-30. Inevitably, the mentioned NRW sharp increase–decrease behavior in the case of PE-30 resulted in a significant overall range between the minimum and maximum ε′ values (this is once again highlighted in Table 4). While measuring the dielectric constant of Rexolite (also in the frequency range 8.2–12.5 GHz), the authors of [27] acquired an NRW profile very similar to the one observed here for PE-30.

The NRW method, although foundational (given that it was the first of its kind when it was developed), is broadly recognized as an ambiguous method as it necessitates that the phase ambiguity is addressed at each frequency through a comparison of calculated and measured group delay. Additionally, it is commonly observed that the method is not corroborative at frequencies corresponding to integral multiples of one half-wavelength of sample thickness. Due to these undesirable effects, the NRW method was immediately ruled out in our case.

When comparing the PF and EP methods, on the basis of our results, it was found that, although both provide rather stable results, the PF method best addresses the shortcomings of the NRW. Hence, it was chosen as the optimal conversion algorithm, meaning that all the plots that will be presented from this point onwards, were generated from data computed by the PF method.

### 3.2. ‘Covered’ vs. ‘Uncovered’

Figure 5a–c exhibit the uncovered vs covered comparison results for the three different reference materials. Each of the plots corresponds to the dielectric constant as a function of frequency for different materials and the two profiles in each graph pertain to the uncovered and covered calibration scenarios.

At the lower end of the frequency spectrum, the two FR4 profiles lie very close to each other. At these frequencies, the uncovered data is characterized by a high standard deviation (much higher than those corresponding with the covered data). With an increase in frequency, ε′ from the uncovered measurements decreases gradually while that corresponding to the covered measurements remains relatively stable along the spectrum, consistently featuring low standard deviation values. At about 11.4 GHz, the uncovered standard deviation reaches zero and starts increasing gradually again. The intermittently large standard deviation values at the lower frequencies and the instability associated with the uncovered measurements suggest that a scattered number of outliers within the data set (which are rather difficult to detect) might be present.

The behavior observed in the case of FR4 is not exhibited in the plots for PTFE and PE-30. In those cases, the two profiles in each of the plots remain rather stable along the considered frequency spectrum, and the uncovered measurements result in slightly higher ε′ values, particularly in the case of PE-30, although the difference is still very minor (~0.15). The overlapping standard deviation areas and the very small mean profile-separating difference in the PE-30 graph indicate that the difference between the results yielded from the uncovered and covered measurements, respectively, is not statistically significant. Although this overlapping behavior is not observed in the case of PTFE, the mean dielectric constant values are still relatively close to each other.

Given that the measured dielectric constant behavior corresponding to the uncovered measurements is overall very similar to that pertaining to the covered measurements (with the exception of FR4), and that the differences between the corresponding profiles are minimal, it can be concluded that the Kapton tape does not substantially affect the resulting dielectric constant values when an appropriate calibration is performed beforehand. By appropriate calibration, here we mean a calibration whereby the faces of the measurement standards are also covered by Kapton tape.

### 3.3. Soil Compaction Outcomes 

Figure 6a–c present the frequency distribution of ε′ for three different soil types, namely B3, G1, and Ħ9. The graphs pertaining to the other soil types are not presented to avoid redundancy; nonetheless, a complete summary of the dielectric constant values measured for all soil samples is given in Table 5.

Figure 6a–c each correspond with a particular soil type and in turn, the three profiles in each graph are associated with the corresponding density levels achieved. In this case, all profiles appear to feature a very similar trend. In some cases, as one can also deduce from Table 5, ε′ remains constant throughout the entire frequency spectrum (e.g., B3 Low Density). In the cases when ε′ does not remain constant, the observed variations are not statistically significant. 

It can be noted that there exists a clear relationship between density ($\rho_{\mathrm{low}}$ and $\rho_{\mathrm{high}}$) and the dielectric constant, ε′: the latter increases for higher densities. Such an occurrence is somewhat expected as compressed soil contains fewer pore spaces (air has a low ε′), and more soil particles and water molecules. The unequal spacing between the profiles in the respective graphs originates from the fact that the difference between the low and medium densities is not exactly equal to that between the medium and high densities. To give an example, in the case of B3, the medium density is more similar to the low density than it is to the high density. Consequently, the low- and medium-density profiles lay closer to each other. The best distribution, in terms of spacing between the respective profiles, is exhibited in the case of Ħ9, i.e., Figure 6c. In fact, in this case, the difference between the respective densities (low-medium and medium-high) is very similar (approximately 0.200 ${g\;\mathrm{cm}}^{-3}$). 

Figure 7a,b present the variation in ε′ as a function of frequency for all six soil samples at low and high density, respectively. From these graphs, it can be noted that the ‘low density’ and ‘high density’ values are not exactly the same for all soil samples and that the profile sequence in the low-density graph is not exactly the same as that in the high-density graph. At a low density, Ħ5 and Ħ9 yielded the highest and lowest ε′ values, respectively. Conversely, at high density, it was G1 that yielded the highest values (with Ħ9 also resulting in the lowest ε′ values). Taking the high-density case as an example, the position of the Ħ9 profile can be explained by the fact that the high density achieved for this same soil sample was low in comparison to that achieved for the other samples (difference of ~0.1 ${g\;\mathrm{cm}}^{-3}$), and it is quite clear from Figure 6 that the dielectric constant increases with soil compaction. Conversely, the high dielectric constant values of the G1 profile do not seem to be related directly to the density. This provided that, for the majority of the other samples (naturally excluding Ħ9), a higher density was achieved (even if the differences are quite minute). The observed difference in patterns between the low- and high-density cases may be explained by the fact that, when soil is compressed, its physical properties are in a way being altered, simultaneously affecting its electrical properties such as the dielectric constant. The low-density graph, for which no compaction was involved, does not shed much light on the relationship between soil physical properties and the dielectric constant at the frequency range considered. In [46], the authors tried to implement a soil dielectric measurement method that is not affected by the physical properties of the soil, the latter proving to be somewhat sensitive to work with.

### 3.4. Is ε′ a Viable SWC Indicator for Soils of Malta and How Are the Two Related?

Figure 8a,b illustrate the ε′ frequency distributions corresponding with different gravimetric soil water contents, $\theta_{\mathrm{grav}}$, achieved for the B3 and G1 samples, respectively. The corresponding numerical results are summarized in Table 6. Although the low-density results were very similar to the high-density results with regards to trend, the latter naturally featured higher ε′ values. These results suggest that ε′ is highly correlated with $\theta_{\mathrm{grav}}$ (the relationship being a positive one).

The observation above was somewhat expected given the molecular composition of wet soil. Electromagnetically, soils are typically considered as products of three main components: a solid phase constituted by the soil matrix, a gaseous phase (normally air), and a liquid water phase. Occasionally, the latter component is further subdivided into bound and free water. In its liquid phase, water is considered to have a dielectric constant of ~80 (‘approximately’ because the value depends on temperature, electrolyte solution, and signal frequency), while $\varepsilon \prime_{\mathrm{air}}$ is ~1, and that for the solid components ranges from 4 to 16 [9]. Due to this stark contrast, ${\varepsilon \prime }_{\mathrm{soil}}$ is extremely sensitive to changes in the SWC.

The data presented in Figure 8 was used to generate the calibration curves illustrated in Figure 9. $\varepsilon^{\prime }$ for the different saturation percentages at the 9.04 GHz transect were plotted and a polynomial function of the third order, in the form shown below, was fitted:(2)$$\theta_{\mathrm{grav}}=a{\left(\varepsilon^{\prime }\right)}^{3}+b{\left(\varepsilon^{\prime }\right)}^{2}+c\left(\varepsilon^{\prime }\right)+d$$

The terms a, b, c, and d, obtained for the respective profiles are summarized in Table 7. As indicated by the legend in the figures, the scatter points correspond with the measured data while the trend lines are representative of the fitted cubic models. From these figures, it can be observed that the $\varepsilon^{\prime }$ sensitivity to SWC increases for higher saturation values. In fact, $\varepsilon^{\prime }$ increases gradually with moisture content until it reaches a transitional saturation point at which the profiles acquire a more prominent slope. Such an observation was also pointed out by the authors of [15]. In our case, this transition is more evident in the G1 profiles. To assess the accuracy of the fitted models in Figure 9, the $\varepsilon^{\prime }$ values corresponding to each SWC percentage measured by the moisture analyzer were substituted into the relevant generated models, and new SWC values were computed. In fact, the error bars in Figure 9 represent the percentage difference between the SWC values obtained through the moisture analyzer and those computed through the final cubic models. The largest percentage difference in this case was 7.544%, meaning that, overall, the resulting calibration curves can be considered relatively accurate.

Equation (3) represents the relationship between the actual and measured water content for each soil type classification, considering the maximum difference in each category.
(3)Measured SWC=Calculated SWC ±e 

In this case, *Measured SWC* refers to the SWC determined in the laboratory using the moisture analyzer, *Calculated SWC* refers to the SWC determined using the calibration curves presented in Figure 9, and *e* is a constant indicating the maximum difference between the measured and calculated SWC. The values of *e* for the different categories are summarized in Table 8.

The imaginary component of the complex permittivity, i.e., the loss factor, $\varepsilon^{\prime \prime }$, for wet soil originates from the relaxation behavior of water molecules and the presence of ions that improve the overall conductivity. A summary of the $\varepsilon^{\prime \prime }$ results is presented in Table 9. From this table, one can deduce how $\varepsilon^{\prime \prime }$ increases with increasing density and SWC. In most cases, as expected, $\varepsilon^{\prime \prime }$ increases slightly with frequency as a consequence of the relaxation mechanisms. This increase corresponds to the small decrease in $\varepsilon^{\prime }$ exhibited by the majority of profiles in Figure 8. Overall, however, the $\varepsilon^{\prime \prime }$ values are smaller than those of $\varepsilon^{\prime }$.

## 4. Summary and Conclusions

The primary aims of this study were to determine whether the dielectric constant, $\varepsilon^{\prime }$, can be considered as a viable SWC indicator when it comes to Maltese soils, and to generate two calibration curves for two Maltese soil types, relating directly, the gravimetric soil water content, $\theta_{\mathrm{grav}}$, as a percentage, and $\varepsilon^{\prime }$. The analysis was carried out in the range 8.2–12.4 GHz (X-band) using a setup comprising the two-port, two-path Rohde and Schwarz ZVA-50 10 MHz-50 GHz Vector Network Analyzer (VNA) and a rectangular waveguide system.

Before the soil moisture measurements, a set of initial measurements were carried out to validate the proposed experimental setup and to choose the ideal S-parameters conversion algorithm. During these measurements, three reference dielectric materials (FR4, PTFE, and PE-30), and three conversion algorithms (Nicolson–Ross–Weir, Epsilon Precision, and Polynomial Fit), were tested. The main conclusions were that: (i) the resulting $\varepsilon^{\prime }$ for the reference materials were similar to those presented in the available literature; (ii) a layer of Kapton tape on either side of the sample holder, introduced to keep the soil samples in place, does not substantially affect $\varepsilon^{\prime }$; and (iii) the Polynomial Fit algorithm provides the best $\varepsilon^{\prime }$ values.

During the soil measurements, two primary soil variables were focused on: density and moisture content. During analysis of the former, the given soil samples were left at their original saturation levels and compressed differently, while for the latter, two soil types were saturated with different water levels. In both cases, positive correlations were observed. Hence, it was concluded that $\varepsilon^{\prime }$ can be considered a reasonable SWC indicator for the soils of Malta at low and high density. The correlation between $\varepsilon^{\prime }$ and SWC was quantified mathematically by a set of third-order polynomial functions that were fitted to the measured data. In the future, our findings should facilitate the development process of minimally invasive and low-cost MW applications for localized SWC sensing, such as the one outlined in [23]. Thus, they can serve as the basis for numerical and analytical models to retrieve the SWC from measurements of the dielectric constant. Such systems are key when it comes to water conservation in agricultural practices. However, it should be noted that a clear relationship between the soil texture and $\varepsilon^{\prime }$ could not be determined at this stage.

## Figures and Tables

**Figure 1 sensors-23-05357-f001:**
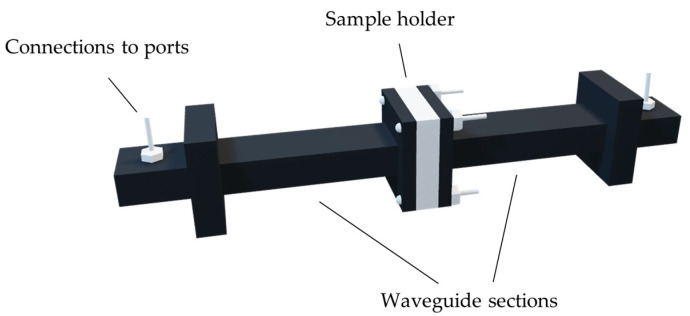
A simplified three-dimensional model of the two waveguide sections after being connected together, with the sample holder in between and the bolt screws inserted accordingly.

**Figure 2 sensors-23-05357-f002:**
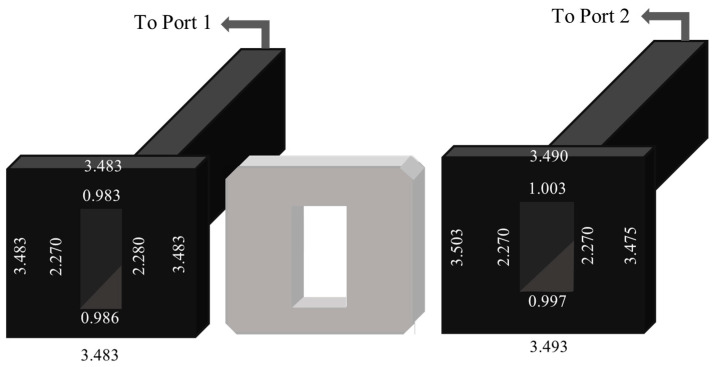
Schematic diagram of the two rectangular waveguide sections employed in our study (left and right) and an empty sample holder (center). It should be noted that these are not drawn to scale and are only meant to give a general idea of the respective dimensions. All dimensions are in centimeters.

**Figure 3 sensors-23-05357-f003:**
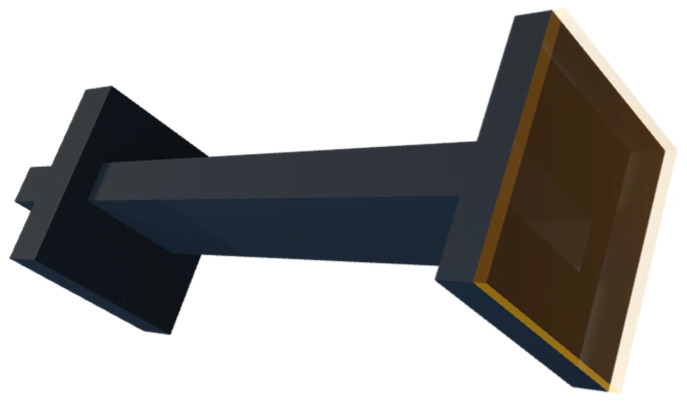
Layer of Kapton tape attached to the end of a waveguide section. The layer thickness shown in the figure is not to scale.

**Figure 4 sensors-23-05357-f004:**
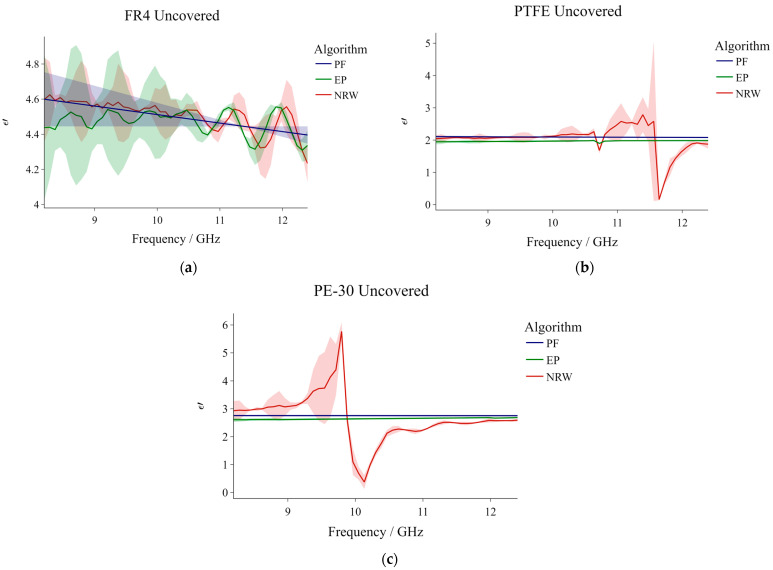
Different conversion algorithm dielectric constant profiles for (**a**) FR4; (**b**) PTFE; and (**c**) PE-30. The results shown are for ‘uncovered’ samples.

**Figure 5 sensors-23-05357-f005:**
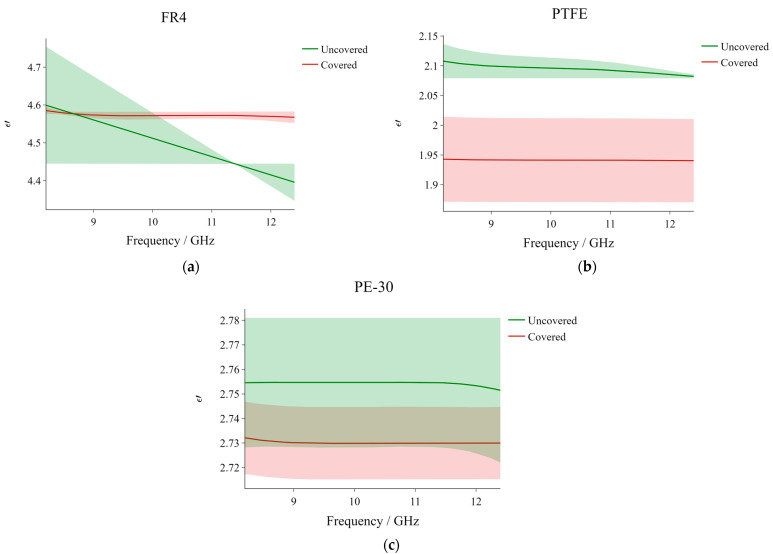
Dielectric constant vs frequency profiles comparing the ‘uncovered’ and ‘covered’ results for (**a**) FR4; (**b**) PTFE; and (**c**) PE-30.

**Figure 6 sensors-23-05357-f006:**
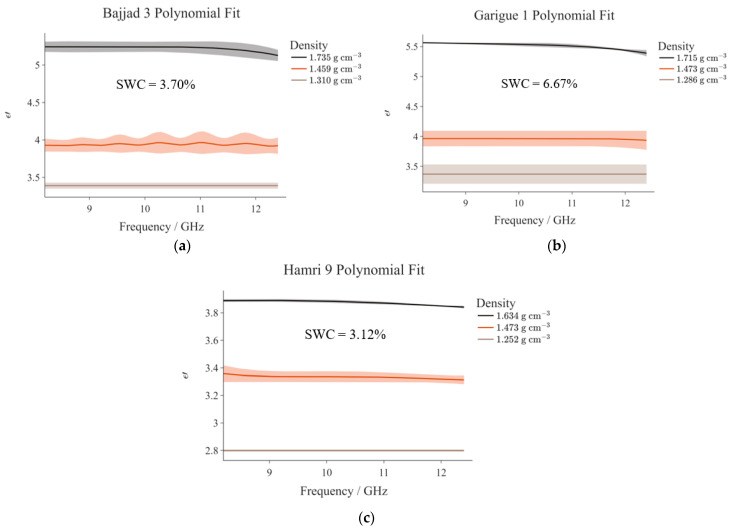
ε′ frequency distribution for (**a**) B3; (**b**) G1; and (**c**) Ħ9. The three profiles in each graph correspond with different soil densities achieved by manual compaction.

**Figure 7 sensors-23-05357-f007:**
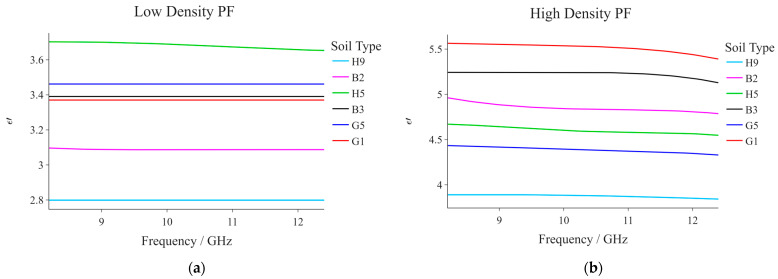
Dielectric constant as a function of frequency for all given soil samples at (**a**) low, and (**b**) high density, respectively.

**Figure 8 sensors-23-05357-f008:**
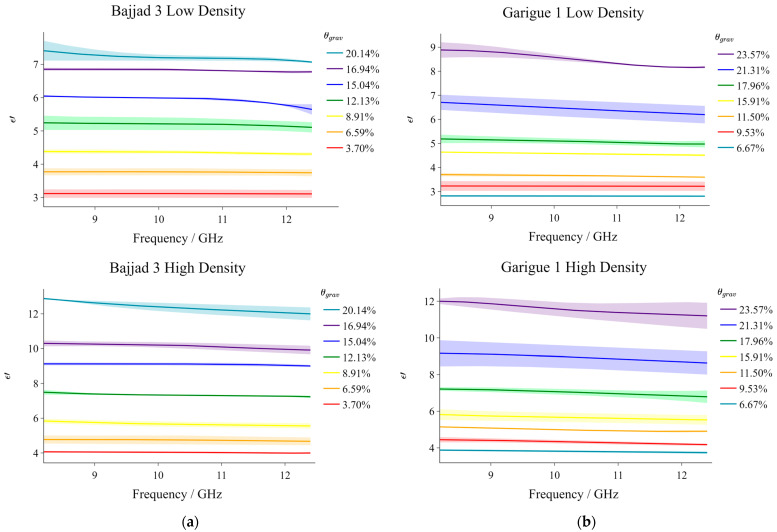
Moisture results at low and high density, respectively for (**a**) Bajjad 3, and (**b**) Garigue 1.

**Figure 9 sensors-23-05357-f009:**
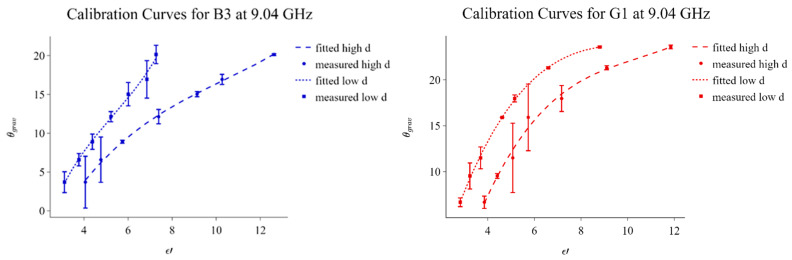
Calibration curves for B3 (**left**) and G1 (**right**) obtained from data taken at a 9.04 GHz transect. While the scatter points correspond with the measured data, the trendlines represent the fitted cubic models. The error bars are associated with the percentage difference between the SWC values measured by the moisture analyzer and those obtained after substituting $\varepsilon^{\prime }$ values into the relevant models.

**Table 1 sensors-23-05357-t001:** Characteristics of the soil samples used in this study.

Soil	Lang [29] Kubiena [34]	WRBS	Clay (%)	Silt (%)	Sand (%)	OM (%)	EC_(SE)_ (μS cm^−1^)
Bajjad 3	Xerorendzina San Biagio Series	Calcisol	23	55	22	1.9	4087
Ħamri 5	Terra Rossa Tas-Sigra Series	Luvisol	34	46	20	2.8	3503
Ħamri 9	Terra Rossa Tas-Sigra Series	Luvisol	10	49	41	3.9	2901
Bajjad 2	Xerorendzina San Biagio Series	Calcisol	26	39	35	1.9	2253
Garigue 1	Terra Rossa Tax-Xaghra Series	Leptosol	23	53	24	2.2	1765
Garigue 5	Terra Rossa Tax-Xaghra Series	Leptosol	24	54	18	1.9	1772

**Table 2 sensors-23-05357-t002:** Density values achieved for different soil types and calibrations.

Soil	Low Density $\left(g\;{\mathrm{cm}}^{-3}\right)$		Medium Density $\left(g\;cm^{-3}\right)$		High Density $\left(g\;cm^{-3}\right)$	
	Cal 1	Cal 2	Cal 1	Cal 2	Cal 1	Cal 2
B3	1.3096	1.3100	1.4587	1.4591	1.7330	1.7343
Ħ5	1.3987	1.4134	1.5196	1.5278	1.7604	1.7713
Ħ9	1.2443	1.2591	1.4639	1.4830	1.6348	1.6330
B2	1.2417	1.2409	1.5047	1.5017	1.7443	1.7152
G1	1.2843	1.2883	1.4743	1.4713	1.7148	1.7148
G5	1.4330	1.4235	1.5443	1.5478	1.7287	1.7365

**Table 3 sensors-23-05357-t003:** Complete set of gravimetric SWC values acquired from the heating moisture analyzer and corresponding to the different levels of added water, together with the densities achieved.

Soil	$\theta_{\mathrm{grav}}\left(\%\right)$	$\rho_{\mathrm{low}}\;$	$\rho_{\mathrm{high}}\;$
	±0.01	$\left(g\;{\mathrm{cm}}^{-3}\right)$	$\left(g\;{\mathrm{cm}}^{-3}\right)$
B3	3.70	1.152	1.394
	6.59	1.144	1.416
	8.91	1.168	1.447
	12.13	1.172	1.484
	15.04	1.144	1.534
	16.94	1.162	1.514
	20.14	1.170	1.562
G1	6.67	1.387	1.593
	9.53	1.379	1.606
	11.71	1.403	1.666
	15.91	1.378	1.571
	17.96	1.372	1.598
	21.31	1.378	1.625
	23.57	1.365	1.605

**Table 4 sensors-23-05357-t004:** A summary of the dielectric constant (ε′ ) results for the ‘uncovered’ reference materials.

Conversion Method	$\bar{\varepsilon \prime }$		
	FR4	PTFE	PE-30
NRW	4.5084 ± 0.0854	2.0405 ± 0.4290	2.6526 ± 0.8774
EP	4.4685 ± 0.0658	1.9692 ± 0.0162	2.6466 ± 0.0253
PF	4.4975 ± 0.0599	2.0946 ± 0.0062	2.7543 ± 0.0007

**Table 5 sensors-23-05357-t005:** A summary of the density results for the six soil samples, in terms of the average dielectric constant, $\bar{\varepsilon \prime }$ (obtained from all frequencies). In all cases, no additional water was added.

Soil	$\bar{\varepsilon \prime }\;$		
	$\rho_{\mathrm{low}}$	$\rho_{\mathrm{medium}}$	$\rho_{\mathrm{high}}$
B3	3.3908 ± 0.0000	3.9397 ± 0.0123	5.2256 ± 0.0298
Ħ5	3.6823 ± 0.0166	3.6993 ± 0.0036	4.6026 ± 0.0348
Ħ9	2.7988 ± 0.0000	3.3335 ± 0.0095	3.8760 ± 0.0154
B2	3.0880 ± 0.0021	3.8255 ± 0.0149	4.8503 ± 0.0416
G1	3.3703 ± 0.0000	3.9614 ± 0.0066	5.5145 ± 0.0458
G5	3.4617 ± 0.0001	3.6504 ± 0.0022	4.3876 ± 0.0286

**Table 6 sensors-23-05357-t006:** Numerical moisture results summary for dielectric constant.

		$\rho_{\mathrm{low}}$		$\rho_{\mathrm{high}}$	
Soil	${\bar{\theta }}_{grav}\left(\%\right)$	$\bar{\varepsilon \prime }$	ε′ Range	$\bar{\varepsilon \prime }$	ε′ Range
	±0.01				
B3	3.70	3.113	3.105–3.118	4.035	3.991–4.065
	6.59	3.764	3.743–3.773	4.737	4.670–4.772
	8.91	4.353	4.304–4.381	5.669	5.554–5.839
	12.13	5.200	5.107–5.244	7.343	7.235–7.494
	15.04	5.944	5.647–6.047	9.105	9.007–9.131
	16.94	6.826	6.772–6.855	10.149	9.915–10.299
	20.14	7.215	7.068–7.410	12.374	11.995–12.878
G1	6.67	2.818	2.807–2.820	3.809	3.740–3.878
	9.53	3.229	3.221–3.232	4.316	4.176–4.445
	11.71	3.661	3.598–3.703	4.989	4.898–5.146
	15.91	4.575	4.511–4.639	5.657	5.532–5.821
	17.96	5.083	4.974–5.188	7.030	6.791–7.214
	21.31	6.446	6.198–6.708	8.935	8.634–9.160
	23.57	8.507	8.160–8.886	11.566	11.202–11.998

**Table 7 sensors-23-05357-t007:** Summary of the terms in the respective cubic polynomial equations representing the fitted curves.

Specimen	a	b	c	d	$R^{2}$
B3 Low Density	0.1342	−2.1454	14.7100	25.4640	0.9942
B3 High Density	0.0181	−0.5560	7.0638	−16.7840	0.9985
G1 Low Density	0.0349	−1.1173	11.9450	−18.8490	0.9993
G1 High Density	0.0376	−1.1699	12.9520	−28.1050	0.9900

**Table 8 sensors-23-05357-t008:** Summary of the values of e representing the maximum difference between the measured and calculated SWC for the different soil types.

Soil Type	*e*
B3 Low Density	2.417
B3 High Density	3.334
G1 Low Density	1.413
G1 High Density	3.772

**Table 9 sensors-23-05357-t009:** Summary of the moisture results for loss factor.

		$\rho_{\mathrm{low}}$		$\rho_{\mathrm{high}}$	
Soil	${\bar{\theta }}_{\mathrm{grav}}\left(\%\right)$	$\bar{\varepsilon \prime \prime }$	ε″ Range	$\bar{\varepsilon \prime \prime }$	ε″ Range
	±0.01				
B3	3.70	0.154	0.107–0.311	0.206	0.094–0.271
	6.59	0.433	0.407–0.514	0.625	0.497–0.729
	8.91	0.591	0.534–0.651	0.846	0.757–0.946
	12.13	0.953	0.857–1.157	1.274	1.142–1.484
	15.04	1.643	1.296–2.195	1.817	1.737–2.114
	16.94	1.566	0.962–1.869	2.628	1.421–3.154
	20.14	2.228	2.135–2.415	2.841	2.201–3.041
G1	6.67	0.308	0.221–0.430	0.349	0.269–0.425
	9.53	0.449	0.350–0.559	0.541	0.520–0.592
	11.71	0.497	0.439–0.587	0.621	0.464–0.722
	15.91	0.541	0.324–0.641	1.192	1.030–1.248
	17.96	1.047	0.988–1.186	1.585	1.406–1.771
	21.31	1.663	1.516–1.716	2.071	1.510–2.171
	23.57	2.324	2.163–2.561	2.637	2.189–3.192

## Data Availability

Data is currently not available.

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
