# Peer review of "Broadband Measurements of Soil Complex Permittivity"

_sensors, 2023, doi:10.3390/s23115357_

Round 1

Reviewer 1 Report

The authors of the manuscript "Broadband measurements of soil complex permittivity" experimentally study the dielectric properties of several types of Maltese soils with different moisture levels. I could have recommended this manuscript for publication, certainly with some reservations, if the authors had succeded to demonstrate the practical implementability of their work. Unfortunately, this has not been done in the current version. The frequency range chosen for the dielectric measurements is very infelicitous for the field studies because the penetration depth at these frequencies is no greater than a few centimeters, according to the reported permittivity data. This means that the applicability of time domain reflectometry for sensing soil water content, which was hinted at in Conclusions, is extremely limited in depth range and totally unsuitable for the field task. (I realize it could be accomplished upon core sampling, but this makes things a lot more difficult and is not pertinent to remote sensing.) Moreover, the authors themselves underline in Abstract that "a statistically significant relation between soil physical properties and the dielectric constant could not be determined at this stage" as supported by the data from Figure 8. By the same token, the statement "It was hence concluded that ?′ can be considered as a reasonable SWC indicator for Maltese soils" in Conclusions does not seem tenable.

It is worthwhile to note that at lower frequencies, the imaginary part of the dielectric permittivity may be far more sensitive towards conductive media, e.g. soil water. Along with the higher penetrability of lower frequencies, this option seems much more viable in the field work.

Soil water sensing is a long-standing problem that has been under research since about the 1960-s, the radio instrumentation heyday. Given this, I would expect that the authors present, in Introduction, some kind of a short overview on advances in this area and highlight, in the framework of Discussion, the advantages of their method compared with previous ones. The authors should somehow justify their choice of method.

Apart from these major concerns, there are other issues regarding the manuscript contents.

It is not clear why the moisture content and other properties were analyzed only in two soil samples (B1 and G1).

In Figure 5, the authors present the results of the dielectric measurements for three soil samples. It is not clear which moisture levels they correspond to. The authors specify in Table 5 that no additional water was added. What are then the intrinsic soil moisture levels? It appears that thay cannot be completely dry as the dielectric constant increases significantly with the compaction as noted also by the authors in L. 530.

There are two unexpalined abbreviations (MUT in L.107 and TRL in L.177.)

L. 621: the word "gradient" is not correct here as "gradient" means the spatial coordinate derivative. "Slope" is a more appropriate word.

Reviewer 2 Report

Soil water content (SWC) is important in agricultural production to optimize crop yield and water use. Broadband measurements of soil complex permittivity were carried out in this study, using an experimental setup comprising of a two-port Vector Network Analyzer (VNA) connected to a rectangular waveguide system. Three kinds of standard materials were measured to calibrated the test method. Three conversion algorithms (NRW, EP, and PF) were also tested and compared. In this paper, calibration of test system and comparison of measurement method are carried out, followed by measurement of a variety of soil samples, and discussion of measurement results are presented. This paper provides a feasible idea to carry out soil water content estimation in agricultural production. Meanwhile, the manuscript still needs to be improved.

1.     The innovation of the test method is not adequately described in this paper, the author is suggested to improve it in the manuscript.

2.     The abstract of the manuscript needs to be revised to reflect the advantages and significance of the work to meet the requirements of journal.

3.     The structure of the manuscript is more like a research report, which may need more in-depth analysis and discussion.

4.     The dimension parameters of the waveguide measurement system need to be provided.

Round 2

Reviewer 1 Report

The authors have addressed some of my comments and suggestions from the previous review. Unfortunately, a number of the answers given do not seem satisfactory. The use of the dielectric constant as the indicator of SWC in Maltese soils could be admitted as the legitimate purpose of the work. However, the dielectric constant cannot be regarded as the "reasonable indicator" of water content in real soils, as it follows from Figure 9 and contrary to what has been concluded, because it was shown to depend on both the SWC and soil compaction.

In the response, the authors write: "We understand our shortcomings in not specifying this in the manuscript, and have now implemented the necessary modifications to highlight this point." However, I have not found any indications of the method limitations in the revision.

"During density investigations, no additional water was added to the soils. The latter were left with their intrinsic water contents which we have now also included in the density plots." - Which plots? Where is the intrinsic water content indicated?

In addition, the authors should correct the citation style in Table 1 and specify in M&M what follows after "±" in the results (SD, SE or CI).

Overall, I would recommend that the authors should pay more attention to reviewer's comments and respect the reader by facilitating her/his comprehension of the material presented.

Reviewer 2 Report

The author added relevant contents in the revised version and carefully answered the reviewer's questions, but there are still a few issues to be noted.

1.     The “figures 5” may need to be changed to “figure 6” in line 595.

2.     According to the manuscript and the response letter, the relationship between complex permittivity and SWC has been reported in several studies. In this study, authors only replaced the soil samples and then carried out the same research as the previous ones. There are no new findings in this study. For example, the relationship between soil type and the complex permittivity has not been studied. Therefore, this article lacks innovative contributions in research.

3.     It is mentioned in the paper that both soil density and water content will affect the complex permittivity of soil, so how to distinguish the effects of the two on the complex permittivity may be explored.

4.     If the minimally invasive, low cost MW systems for localized (a few centimetres away from the sensors) SWC sensing mentioned in the response letter has been designed  according to the measured results, it will also be a more meaningful work.

Round 3

Reviewer 1 Report

In the revision. the authors have taken into account my comments and answered my questions. I feel their corrections are sufficient to recommend the manuscript for publication.

Author Response

Noted with thanks.